# Dual-Parameter Sensor for Temperature and Strain Measurement Based on Antiresonance Effect and Few-Mode Fiber

**Shaocui Jiang, Peng Yang, Zenghui Wang, Yujuan Zhang, Wangge Bao and Baojin Peng \***

Key Laboratory of Optical Information Detecting and Display Technology, Zhejiang Normal University, Jinhua 321004, China; jiangsc0428@zjnu.edu.cn (S.J.); 15958968873@163.com (P.Y.); wangzenghui@zjnu.edu.cn (Z.W.); 15072008269@163.com (Y.Z.); baowangge@163.com (W.B.)
\* Correspondence: jhpbj@zjnu.cn

**Abstract:** A simple and novel hybrid interferometer based on the antiresonance (AR) effect and Mach–Zehnder interference (MZI), which enables simultaneous measurement of temperature and strain, is proposed and investigated. The sensor is made by cascading a 30 cm section of a few-mode fiber (FMF) and a 3.376 mm hollow-core fiber (HCF) through a single-mode fiber (SMF). The FMF and SMF are fused without misalignment to excite two stable modes, thereby forming a Mach–Zehnder interferometer. Concurrently, the introduction of HCF can effectively excite the AR effect, which is manifested in the transmission spectrum as two different dips at the same time caused by the difference in the two physical mechanisms, showing diverse responses to both external temperature and strain. This difference can be used to construct a cross-coefficient matrix to implement the simultaneous measurement of temperature and strain. The experimental results demonstrate that the AR effect and MZI correspond to strain sensitivities of –0.87 and –2.29 pm/$\mu\varepsilon$, respectively, and temperature sensitivities of 15.68 and –13.93 pm/°C, respectively. Furthermore, the sensor is also tested for repeatability, and the results show that it has good repeatability and great potential in sensing applications.

**Keywords:** antiresonance effect; few-mode fiber; simultaneous measurement; temperature; strain

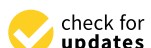



## 1. Introduction

In recent years, optical fiber sensors have attracted great interest owing to their compact structure, high sensitivity, flexibility, and high immunity to electromagnetic interference [1–3]. Various types of fiber optic sensors have been widely used in many fields, such as aerospace, civil engineering, and structural health monitoring. In general, in practical applications, temperature variations can introduce additional errors into the sensing results. To avoid this issue, special optical fibers containing temperature-compensating elements are usually designed in the sensing system [4–7], but this design can make the system quite complicated. For instance, several sensing structures combining fiber Bragg grating (FBG) [8–10], long period fiber grating (LPG), and Fabry–Perot interferometer (FPI) [11–13] have been proposed. Additionally, researchers have been devoted to studying low-temperature cross-talk high-sensitivity strain sensors. For example, Zhao et al. designed an air-bubble-based FPI in a tapered hollow-core fiber (HCF), achieving a high strain sensitivity of 8.62 pm/$\mu\varepsilon$. The following year, the same authors developed a dual S-tapered fiber strain sensor with a strain sensitivity of 6.63 pm/$\mu\varepsilon$. Both of these tapered structures exhibit low temperature sensitivity, thereby eliminating the need to consider temperature-induced cross-talk [14–16]. Another approach is to use a cascade structure to achieve simultaneous measurement of temperature and the desired physical parameters. In this case, multiple fiber hybrid cascade structures have been reported to achieve simultaneous measurement of refractive index and temperature [17], strain and temperature [18], and

curvature and temperature [19]. Of these, the simultaneous measurement of temperature and strain is of significant investigative value in areas such as environmental monitoring and structural engineering. Therefore, considering the requirements of modern industry, it is necessary to design a new structure to achieve simultaneous measurement of strain and temperature.

Fiber optic sensors based on an antiresonant reflecting optical waveguide (ARROW), including photonic crystal fibers for refractive index sensing [20], negative curvature hollow-core fibers (HCFs) [21], single-hole twin-suspended core fibers (SHTSCFs) [22], and HCFs for barometric, level, and temperature sensing [23–25], have been intensively studied. However, most of these methods can only achieve measurements of individual parameters by directly tracking the shift in the transmission spectrum. Consequently, temperature crosstalk in complex environments cannot be avoided. ARROW combination with other interferometric mechanisms has been reported extensively in the sensing of several parameters. For example, Gui et al. used a 3 mm-long SHTSCF to simultaneously excite an antiresonance (AR) effect and Mach–Zehnder interference (MZI) to achieve simultaneous curvature and temperature sensing through intensity demodulation and wavelength demodulation [26]. Zuo et al. combined two segments of a multimode fiber spliced into a conventional SHS structure and fabricated a sensor structure capable of exciting both the AR effect and MZI guidance mechanisms through adequate simulations and experiments, thus achieving simultaneous temperature and strain measurements [27]. Nan et al. proposed a hybrid sensor for simultaneous measurement of three parameters that consists of a segment of HCF fused between an air bubble and an uptaper. The FPI, MZI, and the AR effect were excited simultaneously. Transverse loads could be measured by demodulating the reflection spectrum of the FPI, and curvature and temperature could be measured simultaneously by demodulating the wavelength and intensity [28]. To introduce other interferometric mechanisms to achieve multiparameter measurements, the option is to either modify conventional SHS structures, such as an air bubble and uptaper, or choose expensive antiresonant fibers, which not only complicates the operation but also reduces its reproducibility.

In this paper, we propose a hybrid fiber optic sensor based on MZI and the AR effect to achieve simultaneous measurement of strain and temperature. Given that the two sensing mechanisms respond differently to temperature and strain, the drift of the HCF and few-mode fiber (FMF) transmission spectra can be monitored in real time by using a spectral analyzer to establish a two-parameter coupling matrix. Its temperature sensitivity reaches 15.68 pm/°C and its strain sensitivity reaches $-2.29$ pm/$\mu\varepsilon$. At the same time, the cascade structure of the sensor is simple to produce, and it presents good repeatability and has many advantages; thus, it has a broad application prospect in engineering fields such as safety monitoring.

## 2. Principle

### 2.1. Principle of MZI

The schematic of the proposed sensor, which consists of single-mode fiber (SMF)-FMF-SMF-HCF-SMF (SFSHS) cascade, is shown in Figure 1. Light enters from the input SMF. When passing through the fusion points of SMF1 and FMF, the light transmitted in SMF1 will couple into the FMF core and excite higher-order modes owing to the difference in their mode fields. Figure 2a shows the corresponding mode waveguide dispersion diagrams in the FMF. Four modes in the FMF, which are $LP_{01}$, $LP_{11}$, $LP_{21}$, and $LP_{02}$ modes, are used in this experiment. Although FMF supports the transmission of four modes, the number and intensity of the actual excited modes in FMF are also related to the excitation method, which is commonly used for staggered, tapered, and spliced multimode fibers or coreless fibers. We analyzed the relationship between the modes excited in FMF and the offset using the beam propagation method (BPM). The core diameters of FMF and SMF are set to 18.5 μm and 9 μm, respectively, and the cladding diameter is 125 μm, and the refractive indices of core and cladding are 1.44979 and 1.44402, respectively. The results are shown in

Figure 2b. When the offset of SMF and FMF fusion splicing is 0, only two modes—$LP_{01}$ and $LP_{02}$—exist in the FMF, and the FSR of the interference spectrum is relatively uniform because there is no interference from other modes. As shown in Figure 2b, when the SMF is fused to the FMF without core-offset, the excited modes are mainly $LP_{01}$ and $LP_{02}$. As the transmission distance increases, the different modes produce phase differences given the difference in transmission constants. When the next fusion point is reached, they are coupled into the fiber core of SMF2 at the output end, thus constituting MZI. The sensing mechanism diagram is shown in Figure 3a, and the total light intensity transmitted to the output SMF end can be expressed as follows [29]:

$$I_{FMF} = I_1 + I_2 + 2\sqrt{I_1 I_2} cos\Delta\varphi \qquad (1)$$

where $I_{FMF}$ denotes the total light intensity through the FMF and $I_1$ and $I_2$ are the intensities of the two modes excited in the FMF. $\Delta\varphi$ is the phase difference between them, which can be expressed as

$$\Delta\varphi = \frac{2\pi\Delta n_{eff}}{\lambda}L_{FMF} \qquad (2)$$

where $\Delta n_{eff}$ is the effective refractive index difference between the fundamental mode and the higher-order mode, $\lambda$ is the wavelength, and $L_{FMF}$ is the length of the FMF. When the phase difference satisfies the condition $\Delta\varphi = (2N+1)\pi$ ($N$ is an integer), a resonance dip will be produced in which the wavelength at the dip can be described as

$$\lambda_{dip} = \frac{2\Delta n_{eff}}{2N+1}L_{FMF} \qquad (3)$$

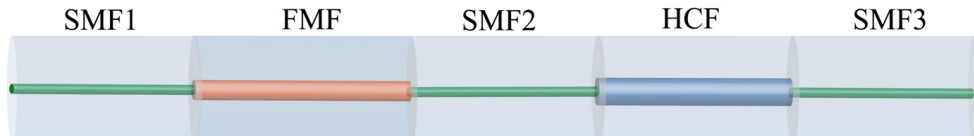

**Figure 1.** Schematic diagram of the proposed sensor.

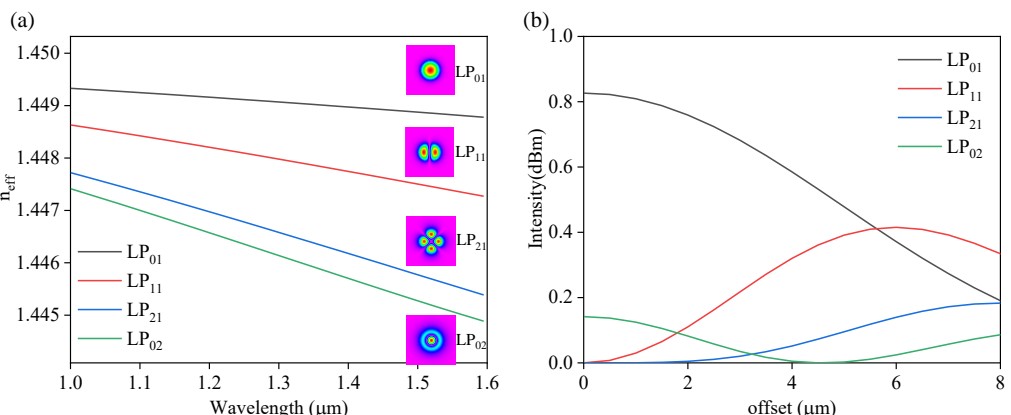

**Figure 2.** (**a**) Effective refractive index versus wavelength for the four modes $LP_{01}$, $LP_{11}$, $LP_{02}$, and $LP_{21}$; (**b**) FMF excited out of the mode versus offset.

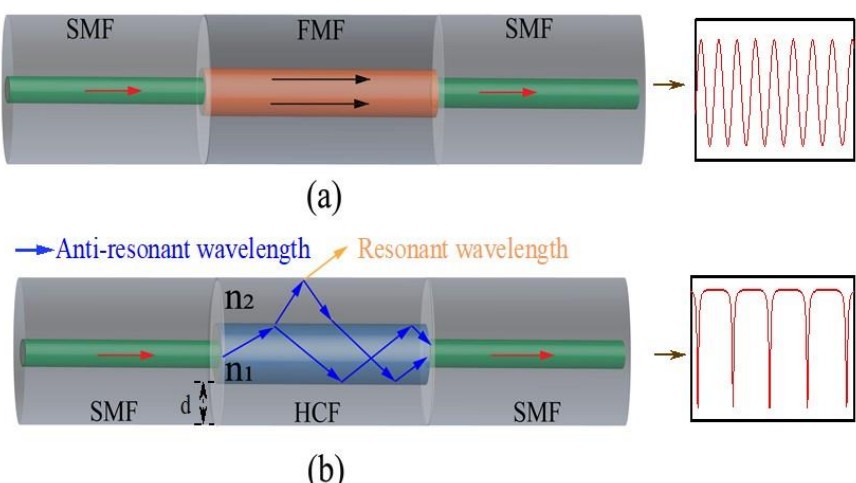

**Figure 3.** (**a**) Sensing schematic of MZI and (**b**) sensing schematic of AR effect.

When the external temperature changes, the change in the length and refractive index difference of the FMF will change the optical range difference between the two paths owing to the thermo-optical effect and thermal expansion effect, which in turn will cause a wavelength drift in the transmission spectrum. Therefore, the relationship between wavelength drift and temperature can be described as [30]

$$\frac{\Delta\lambda}{\lambda} = \left( \frac{1}{L} \cdot \frac{\partial L}{\partial T} + \frac{1}{\Delta n_{eff}} \cdot \frac{\partial \Delta n_{eff}}{\partial T} \right) \Delta T = [\alpha + \xi]\Delta T \tag{4}$$

where $\Delta T$ is the temperature change, $\alpha$ is the thermal expansion coefficient, and $\xi$ is the thermo-optical coefficient.

In addition, when an axial strain is applied, the resonant dip undergoes a wavelength drift because of the photoelastic effect and the change in fiber size, at which time the wavelength drift at the dip can be expressed by the following equation:

$$\frac{\Delta\lambda}{\lambda} = \left[ 1 + \frac{L}{\Delta n_{eff}} \cdot \frac{\partial \Delta n_{eff}}{\partial L} \right] \Delta \varepsilon = (1 + P_e)\Delta \varepsilon \tag{5}$$

where $\Delta \varepsilon$ represents the amount of change in strain and $P_e$ is the effective elastic coefficient.

### 2.2. Principle of the AR Effect

The optical transmission mechanism of the SHS structure can be explained in terms of the AR effect. The beam propagation path of the AR effect is shown in Figure 3b. After the light becomes obliquely incident from the SMF to the HCF, a part of the light transmitted in the air core continues to be reflected, while the other is transmitted into the cladding. The light in the cladding is partly reflected and partly transmitted to the outside world to be lost. The high-refractive-index cladding ($n_2$) can be regarded as an F–P resonant cavity. When the incident light wavelength meets the resonance conditions of the F–P cavity, this part of the light will leak out of the air core and be bound in the high-refractive-index cladding ($n_2$) oscillation, thus showing great loss in the transmission spectrum. When the light wavelength is far from the resonant cavity, the light will be reflected back by the F–P cavity, confined in the low-refractive-index layer ($n_1$), and propagated forward along its axial direction, which shows a very small transmission loss in the transmission spectrum. In accordance with the reflectivity formula of symmetric parallel flat-plate multibeam interference, the transmission spectrum expression of the AR effect can be obtained as follows [31]:

$$I_{AR} = \frac{F sin^2\left(\frac{\varphi}{2}\right)}{1 + F sin^2\left(\frac{\varphi}{2}\right)} \tag{6}$$

where $F$ is the stripe fineness of the interference spectrum and $\varphi$ is the phase difference between two adjacent beams. It can be expressed as

$$\varphi = \frac{4\pi d}{\lambda_m} \sqrt{n_2^2 - n_1^2} \tag{7}$$

where $d$ is the cladding thickness. $\lambda_m$ is the wavelength of the resonant light lost by the AR effect leaking light, and the corresponding expression is [32]

$$\lambda_m = \frac{2d}{m} \sqrt{n_2^2 - n_1^2} \tag{8}$$

where $m$ is the resonance order. Given that the change in refractive index of the air core is negligible compared with that of silica, the effect of temperature on the refractive index of the air core is not considered. The derivative of the resonant wavelength with respect to temperature is given as

$$\frac{\partial \lambda_m(AR)}{\partial T} = \frac{2n_2 d}{m\sqrt{n_2^2 - n_1^2}} \times \frac{\partial n_2}{\partial T} \tag{9}$$

where $\partial n_2 / \partial T$ is the thermo-optical coefficient of the cladding. When the temperature increases, $n_2$ increases, and the resonant wavelength shifts in the long-wavelength direction (red-shift). The resonant wavelength is derived from the strain as

$$\frac{\partial \lambda_m(AR)}{\partial \varepsilon} = \frac{2\sqrt{n_2^2 - n_1^2}}{m} \times \frac{\partial d}{\partial \varepsilon} + \frac{2n_2 d}{m\sqrt{n_2^2 - n_1^2}} \times \frac{\partial n_2}{\partial \varepsilon} \tag{10}$$

where $\partial d / \partial \varepsilon$ is the variation coefficient of cladding thickness with strain and $\partial n_2 / \partial \varepsilon$ represents the variation coefficient of the cladding refractive index with strain.

In consideration of the cascade configuration of SHS and FMF, the final total transmittance output of the sensor is

$$I_{out} = I_{AR} I_{FMF} \tag{11}$$

## 3. Design and Fabrication

The key to achieving simultaneous measurement of strain and temperature by observing the wavelength change is to obtain two distinguishable interference dips. For our proposed sensor, the MZI is formed by accessing a section of the FMF, which is produced by Changfei Fiber Optic Cable Co., Ltd. (Shenzhen, China). It has an inner core diameter of 18.5 μm and a cladding diameter of 125 μm and can stably transmit four modes. The free spectral range of the fused FMF decreases gradually with the increase in the fused FMF length. In the experiments, we chose the appropriate sparse and dense spectral lines for the sensor fabrication, so the FMF used was 30 cm.

The embedded SHS needs reasonable control of the length and inner diameter of the HCF if we want to obtain a significant dip. Therefore, exploring the relevant parameters is necessary. The different states of transmission spectra under the HCF with various inner diameters and the corresponding microscope diagrams are shown in Figure 4a. When the inner diameter of the HCF is 10 and 20 μm, a significant AR can be observed, but it causes excessive loss; when the inner diameter reaches 50 μm and above, the AR effect is less obvious, so the length of the HCF with an inner diameter of 40 μm is explored. As shown in Figure 4b, the longer the length of the HCF is, the more obvious the dip generated by the loss peak.

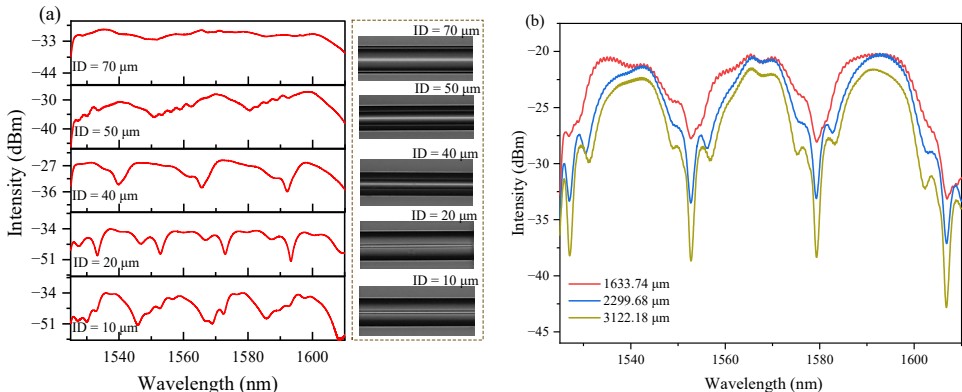

**Figure 4.** (**a**) Transmission spectra and microscope diagrams of HCF with inner diameters of 10, 20, 40, 50, and 70 μm, respectively; (**b**) transmission spectra of HCF with an inner diameter of 40 μm at different lengths.

In accordance with the results of these investigations, we carry out the fabrication of the sensor. Figure 5a–d depict the specific steps. In the first step, the discharge power and discharge time of the fiber fusion splicer are well optimized, and the SMF is discharge-fused to the FMF of determined length. The key parameters are that the predischarge intensity and time equal 198 bit and 50 ms, the main discharge intensity and time 228 bit and 2000 ms. The SHS structure is then prepared using pure quartz HCF (40 μm inner diameter and 125 μm outer diameter), splicing a section of the HCF with a length of 3.376 mm in two sections of SMF. The predischarge intensity and time are 178 bit and 50 ms, the main discharge intensity and time 120 bit and 1500 ms. During the sensor preparation, extra attention is paid to the cutting and fusion of the HCF. In the second step, the two prepared structures are cascaded to form the SHS. Figure 6a shows the transmission spectra obtained from single SMF-FMF-SMF (SFS) and SHS structures. It can be observed that the spectrum obtained from the SFS is a uniform and dense comb-like spectrum. Loss peaks appear at the resonant wavelength position of the SHS structure. Some spurious peaks also appear in the region without resonance, mainly because the thickness of the actual prepared HCF cladding is not absolutely uniform, which leads to the leakage of light from the HCF at other nonresonant frequencies. Figure 6b shows the transmission spectrum of the sensor without strain applied after cascading. The transmission spectrum after cascading appears as a distinguishable dip, which facilitates the multiparameter measurements later on.

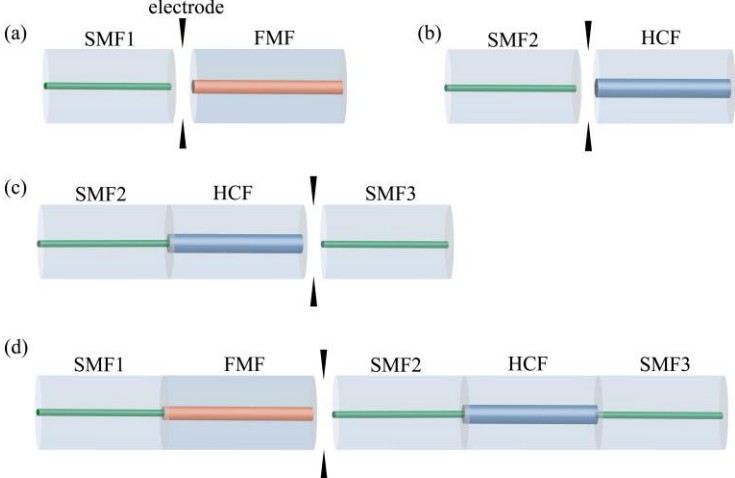

**Figure 5.** The fabrication process diagram of the proposed sensor.(**a**) SMF1 (incident side) is fusion spliced with FMF; (**b**) one end of the HCF is fusion spliced with SMF; (**c**) the other end of the HCF is fusion spliced with SMF3; (**d**) FMF is fusion spliced with SMF2, finally.

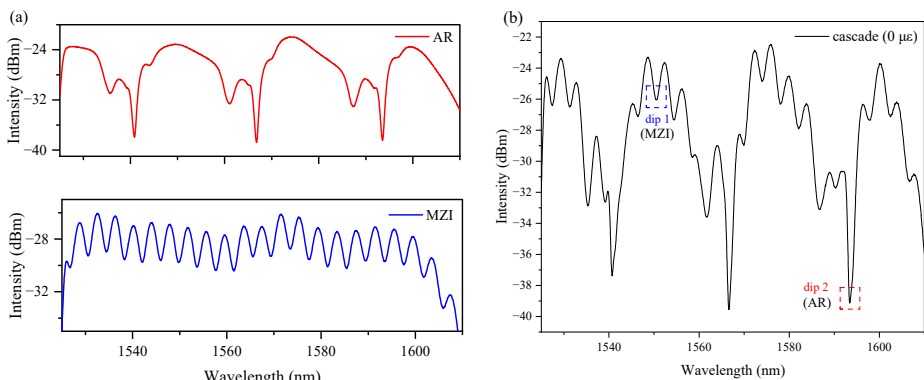

**Figure 6.** (**a**) Transmission spectra of the AR effect and MZI before cascading and (**b**) transmission spectrum of the sensor without applied strain after cascading.

## 4. Experiments and Results

### 4.1. Strain Response Characteristics

The experimental setup shown in Figure 7 was used to study the strain and temperature response. The whole experimental system consisted of an ASE light source (1525–1610 nm, Fiber Lake, China) on the input side, a spectral analyzer (OSA, YOKOGAWA-AQ6375B) on the receiver side with a resolution of 0.02 nm, a displacement stage, and a temperature control box. Both ends of the optical fiber connected to the sensor were fixed to the displacement stage with UV-curable adhesive. One end of the displacement stage was fixed, and the other end was controlled by a computer to change the axial position of the translation stage to gradually increase the length of the optical fiber between the two bonding points. The extension length was increased from 0 to 1400 steps in increments of 100 steps at 0.312 μm per step, and the transmission spectra were recorded immediately at each step (1 με is defined as the tensile force required to stretch a 1 m-long fiber by 1 μm, and 50 με is for a 20 mm-long fiber for each 0.01 mm pull-up). During the strain response test experiments, the room temperature was kept at 25 °C to avoid measurement errors caused by temperature variations.

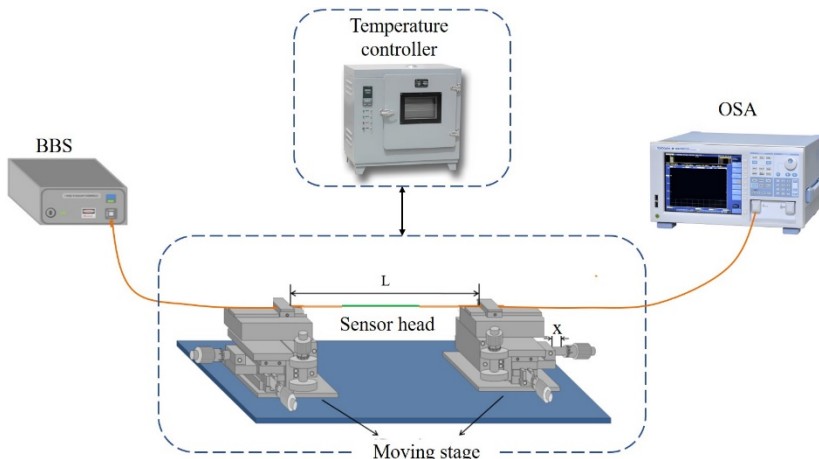

**Figure 7.** Diagram of the experimental setup of the proposed sensor for measuring strain (lower dash box) and temperature (upper dash box) responses.

Figure 8 demonstrates the evolution of the cascade sensor during the increase in strain. With the increase in strain, dip1 induced by MZI and dip2 generated by AR both showed a blue-shift, but the magnitude of the drift was not uniform. The resonant wavelength variation curves of MZI and AR with axial tensile strain were obtained after linear fitting, as shown in Figure 8c. The sensitivities of dip1 and dip2 were −2.29 and −0.87 pm/με, respectively, in the 0–1352 με range, and the linear fitting coefficient went as high as 99.95%.

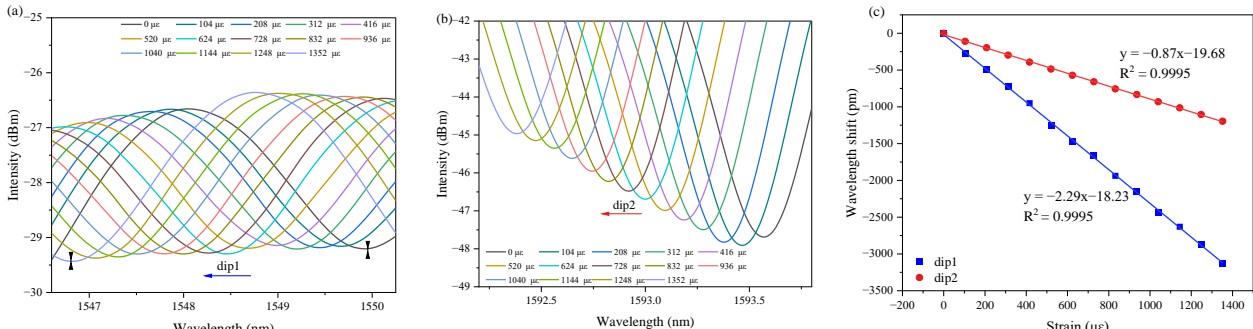

**Figure 8.** Strain response characteristics: (**a**) wavelength drift of dip1 at 0–1352 με strain; (**b**) wavelength drift of dip2 at 0–1352 με strain; (**c**) resonant wavelength of dip1 and dip2 with axial strain.

### 4.2. Temperature Response Characteristics

To characterize the temperature response, the sample of the cascade sensor to be tested was placed in a temperature-controlled thermostat with a temperature range of 10–100 °C. By increasing the temperature in the thermostat by 10 °C each time, the temperature was raised from 30 to 80 °C. Each time the temperature was raised, the data were recorded after 10 min of temperature stabilization to avoid errors caused by temperature instability. The evolution of the transmission spectrum of the cascade sensor in the temperature increase process is shown in Figure 9. As the temperature rose, dip1 generated by MZI was blue-shifted, whereas dip2 generated by AR was red-shifted. The results of the linear fit to the points obtained by dip drift are shown in Figure 9c. The temperature sensitivities of dip1 and dip2 were –13.93 and 15.68 pm/°C, respectively.

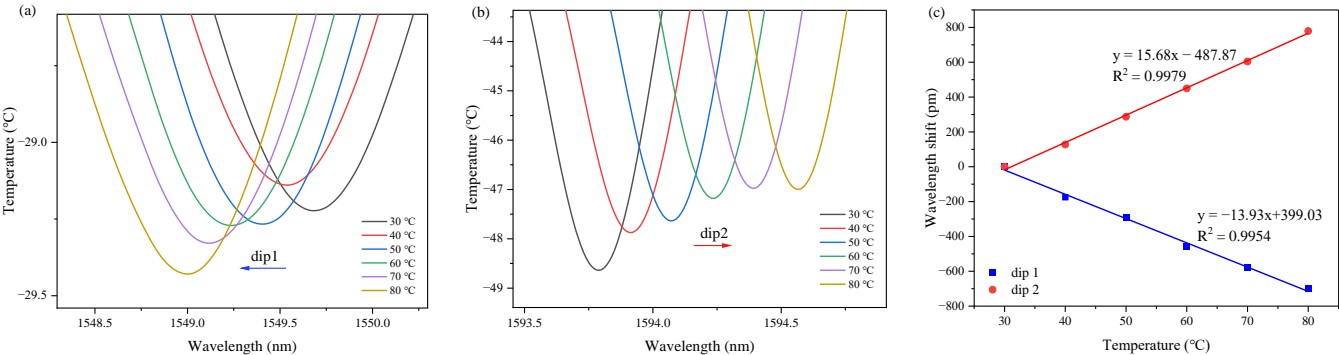

**Figure 9.** Temperature response characteristics: (**a**) evolution of the transmission spectrum of dip1 at a temperature of 30–80 °C; (**b**) evolution of the transmission spectrum of dip2 at a temperature of 30–80 °C; (**c**) variation in the resonant wavelengths of dip1 and dip2 with temperature.

## 5. Discussion

### 5.1. Temperature–Strain Dual-Parameter Demodulation

Fiber optic sensors are usually susceptible to cross sensitivity and to changes in multiple parameters, such as temperature and strain, during sensing measurements, which lead to increased measurement errors. The two sensing mechanisms in the SFSHS sensor studied in this experiment are sensitive to both strain and temperature, but their response sensitivity is somewhat different. Therefore, when both temperature and strain are applied to the sensor, the drift of the resonance peaks dip1 and dip2 due to the change in strain $\Delta\varepsilon$ and temperature $\Delta T$ can be expressed by constructing a sensitivity matrix as [33]

$$\begin{bmatrix} \Delta\lambda_{dip1} \\ \Delta\lambda_{dip2} \end{bmatrix} = \begin{bmatrix} K^{\varepsilon}_{dip1} & K^{T}_{dip1} \\ K^{\varepsilon}_{dip2} & K^{T}_{dip2} \end{bmatrix} \begin{bmatrix} \Delta\varepsilon \\ \Delta T \end{bmatrix} \tag{12}$$

where $\Delta\lambda_{dip1}$ and $\Delta\lambda_{dip2}$ represent the wavelength drift at dip1 and dip2, respectively, $K^{\varepsilon}_{dip1}$ and $K^{T}_{dip1}$ represent the strain and temperature sensitivity coefficients at dip1, and $K^{\varepsilon}_{dip2}$ and $K^{T}_{dip2}$ represent the strain and temperature sensitivity coefficients at dip2 caused by AR. Since the sensitivity coefficients are different, the sensitivity matrix is invertible. The amount of change in temperature and strain derived from Equation (12) can be expressed as

$$\begin{bmatrix} \Delta\varepsilon \\ \Delta T \end{bmatrix} = \frac{1}{|H|} \begin{bmatrix} K^{T}_{dip2} & -K^{T}_{dip1} \\ -K^{\varepsilon}_{dip2} & K^{\varepsilon}_{dip1} \end{bmatrix} \begin{bmatrix} \Delta\lambda_{dip1} \\ \Delta\lambda_{dip2} \end{bmatrix} \tag{13}$$

where $|H| = K^{\varepsilon}_{dip1} \times K^{T}_{dip2} - K^{T}_{dip1} \times K^{\varepsilon}_{dip2}$ is the determinant of the coefficient matrix. For this experimental sensor, the strain and temperature response sensitivities at dip1 and dip2 of $-2.29$ pm/$\mu\varepsilon$, $-0.87$ pm/$\mu\varepsilon$, $-13.93$ pm/$^{\circ}$C, and $15.68$ pm/$^{\circ}$C are sequentially substituted into Equation (13) to obtain

$$\begin{bmatrix} \Delta\varepsilon \\ \Delta T \end{bmatrix} = \frac{1}{-48.03} \begin{bmatrix} 15.68 & 13.93 \\ 0.87 & -2.29 \end{bmatrix} \begin{bmatrix} \Delta\lambda_{dip1} \\ \Delta\lambda_{dip2} \end{bmatrix} \tag{14}$$

combining Equations (12)–(14); the actual strain and temperature in the environment can be measured simultaneously by the drift of the resonance peak of the sensor spectrum.

To demonstrate the practical accuracy of Equation (14) in measurements, a resonant wavelength shift with arbitrary variations in strain and temperature needs to be introduced to the developed sensor. However, due to limitations in the experimental setup, we cannot simultaneously control and vary temperature and strain freely. Therefore, we have followed the method outlined in the referenced article [34] and selected two combinations, namely 60 $^{\circ}$C, 0 $\mu\varepsilon$ and 25 $^{\circ}$C, 520 $\mu\varepsilon$, for analyzing the accuracy of the sensor. The first condition is set as the initial value, while the second condition represents a simultaneous change in temperature and strain. Some of the measurements obtained under this variation are given in Table 1. The measured data under these two conditions are 1549.24 nm, 1594.14 nm, 1548.59 nm and 1593.14 nm respectively. It is known that the values of $\Delta\lambda_{dip1}$ and $\Delta\lambda_{dip2}$ are $-0.65$ nm and $-1.0$ nm, respectively. By substituting these values into Equation (14), we obtain $\Delta\varepsilon$ and $\Delta T$. Therefore, the measured values of strain $\varepsilon_m$ and $T_m$ temperature are 502.2 $\mu\varepsilon$ and 24.1 $^{\circ}$C, respectively, which are relatively close to the actual values with slight errors. The strain error $\varepsilon_{error}$ and $T_{error}$ temperature errors are calculated as 3.4% and 3.6%, respectively. These results indicate that the proposed cascade sensor exhibits good accuracy when measuring strain and temperature simultaneously.

**Table 1.** Simultaneous measurement of $\varepsilon$ and $T$ under certain condition.

| Parameters | Condition (T = 25 $^{\circ}$C, $\varepsilon$ = 520 $\mu\varepsilon$) |
|---|---|
| $\Delta\lambda_{dip1}$ (nm) | $-0.65$ |
| $\Delta\lambda_{dip2}$ (nm) | $-1.0$ |
| Determined data by Equation (14) | $\Delta\varepsilon = +502.2\ \mu\varepsilon$, $\Delta T = -35.9\ ^{\circ}$C |
| Measured $\varepsilon_m$ and $T_m$ | $\varepsilon_m = \varepsilon + \Delta\varepsilon = 502.2\ \mu\varepsilon$ $T_m = T + \Delta T = 24.1\ ^{\circ}$C |
| $\varepsilon_{error} = \lvert\{(\varepsilon_m - \varepsilon)/\varepsilon\}\rvert$ $T_{error} = \lvert\{(T_m - T)/T\}\rvert$ | $\varepsilon_{error} = 0.034$ $T_{error} = 0.036$ |

*5.2. Repeatability Measurement*

To examine whether the sensor is repeatable, we conducted three strain measurements and two temperature experiments using the same sensor and found that the sensor has excellent repeatability for both strain and temperature. As shown in Figure 10a, the sensitivities of three separate measurements of strain were $-2.29$, $-2.26$, and $-2.27$ pm/$\mu\varepsilon$

for dip1 and −0.87, −0.84, and −0.85 pm/με for dip2. The repeatability of temperature measurements was also reliable, as shown in Figure 10b.

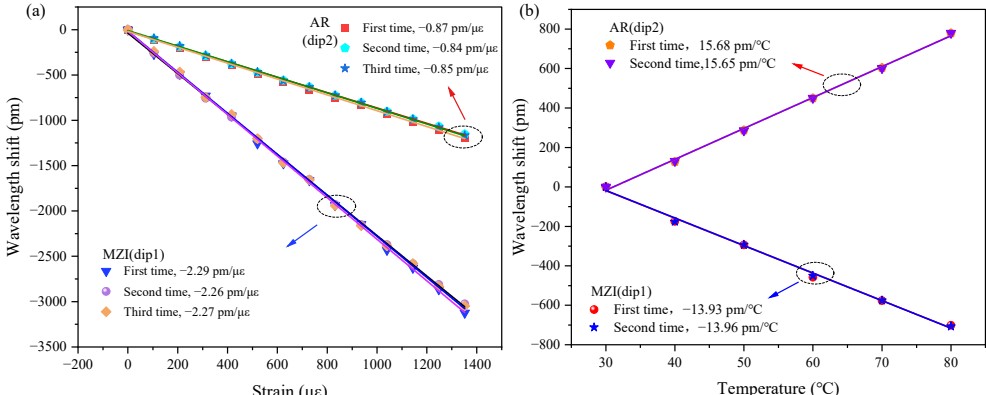

**Figure 10.** (**a**) Verification of the strain repeatability of the proposed sensor by tracking dip1 and dip2 wavelength shifts and (**b**) analysis of the temperature repeatability of the proposed sensor by tracking dip1 and dip2 wavelength drifts.

Table 2 includes a direct comparison of the sensing performance of the sensors described in this study with other sensors for dual parametric measurements of strain and temperature. Compared to some sensing configurations that have been reported [14,15], the strain sensitivity of the sensors proposed in this thesis is relatively low, but they suffer from a certain amount of temperature crosstalk that cannot be resolved at present. Compared to previous work on simultaneous strain and temperature measurements using cascaded FBG and LPG, the incorporation of the AR effect appears to be more convenient and much less costly. In addition, the sensor designed in this paper exhibits a transmission spectrum that is more clearly able to obtain two distinguishable resonance peaks, which is more conducive to simultaneous strain and temperature measurements.

**Table 2.** Performance comparisons of the reported fiber optic strain sensors.

| Configuration | Simultaneous | Strain Sensitivity (pm/με) | Temperature Sensitivity (pm/°C) |
|---|---|---|---|
| FBG and MZI [8] | Yes | −1.83 (0–1000 με) | 46.93 (2.2–80 °C) |
| FMF and FBG [10] | Yes | 0.8778 (0–1000 με) | 9.92 (10–70 °C) |
| LPG and a microsphere [13] | Yes | 0.86 (0–1500 με) | 0.79 (30–80 °C) |
| tapered HCF air-microbubble FPI [14] | No | 8.62 (0–3200 με) | - |
| double S-tapers [15] | No | 6.63 (0–800 με) | - |
| micro-cavity MZI [16] | Yes | 0.02 dB/με (0–300 με) | 0.003 dB/°C (23–28 °C) |
| FPI and ARROW [31] | Yes | 2 (0–1000 με) | 21.11 (30–70 °C) |
| tapered MZI [35] | Yes | 2.7 (0–2100 με) | 1.6 (20–80 °C) |
| ARROW and MZI [36] | Yes | −0.8 (0–1000 με) | 28.5 (0–300 °C) |

**Table 2.** *Cont.*

| Configuration | Simultaneous | Strain Sensitivity (pm/με) | Temperature Sensitivity (pm/°C) |
|---|---|---|---|
| Hollow annular core fiber [37] | Yes | 1.1 (0–1500 με) | 13 (50–500 °C) |
| FMF and ARROW (this work) | Yes | −2.29 (0–1352 με) | 15.68 (30–80 °C) |

## 6. Conclusions

In this paper, we propose and demonstrate a sensor that can realize a hybrid mechanism for simultaneous measurement of temperature and strain. The sensor mainly consists of FMF and HCF cascade: the former can be used to excite MZI, while the latter can excite the AR mechanism. The simulated analysis of FMF and SMF without misalignment fusion determines that the stable $LP_{01}$ and $LP_{02}$ modes form MZI. Different inner diameters and lengths of HCF are explored experimentally, and two easily distinguishable dips are observed in the transmission spectrum of the cascaded sensor. The experimental results show that the spectral lines formed by the two mechanisms have different degrees of response to strain and temperature. Constructing a cross-coefficient matrix between strain and temperature based on these measurements would enable their simultaneous measurement. The proposed sensor has the advantages of easy fabrication and good repeatability, which mean it can realize real-time monitoring of engineering structures and has good prospects for engineering applications. However, the sensitivity is a problem worthy of attention and improvement, and so, using this paper, subsequent experiments will explore the solution to improve that sensitivity.

**Author Contributions:** Conceptualization, S.J. and P.Y.; methodology, S.J. and B.P.; software, P.Y.; validation, S.J., B.P. and Z.W.; writing—original draft preparation, S.J.; writing—review and editing, W.B. and Y.Z.; All authors have read and agreed to the published version of the manuscript.

**Funding:** This research was funded by the Basic Public Welfare Research Project of Zhejiang Province (Grant No. LGG22A040001).

**Institutional Review Board Statement:** Not applicable.

**Informed Consent Statement:** Not applicable.

**Data Availability Statement:** Data sharing not applicable.

**Conflicts of Interest:** The authors declare no conflict of interest.

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
