# Peer review of "Dual-Parameter Sensor for Temperature and Strain Measurement Based on Antiresonance Effect and Few-Mode Fiber"

_photonics, doi:10.3390/photonics10060642_

Round 1

Reviewer 1 Report

The paper describes the operation of a sensor that can measure temperature and strain simultaneously. The authors use a widely used approach of measuring two quantities (in this case - shifts of resonant wavelengths) that depend differently on temperature and strain, followed by solving a system of linear equations.

The experimental part is well described. However, the conclusions about the performance of the sensor are premature.

Suppose I am a potential consumer of the sensor. One of the main characteristics, that is of interest to me, is the accuracy of the parameters measurement. And the article provides absolutely no information about it!

The simplest way to convince the reader that the sensor is working is to carry out appropriate experiments:

- set some values of temperature T and strain ε

- determine resonant wavelengths drifts experimentally

- convert resonant wavelengths drifts into temperature and strain, using Equation (14)

- compare εmeasured and Tmeasured with ε and T, determine the errors

See, for example, the last paragraph in DOI: 10.1364/OL.40.001488.

If for some reason it is not possible to conduct such experiments, the accuracy of the sensor should be estimated theoretically (although this is less convincing). It is very well known that the accuracy of such systems is determined by three factors - non-linearity of responses (how much the dependences in Figures 7c and 8c differ from straight lines), calibration errors (the authors even measured the effect size in section 5.2) and measurement errors (how accurately one can measure wavelengths drifts). And a very important thing is the 'nondegeneracy' of the cross-coefficient matrix (the closer the determinant to zero, the worse the accuracy of the system). Thus, the authors have a complete set of data to evaluate the accuracy of the sensor, but for some reason prefer not to produce it.

And without this data it is absolutely incomprehensible where the sensor can be used in general. Is it good for my application or not?

There are also several minor things to be fixed

1) Introduction section is limited to interference type sensors. As if there are no other sensors (including fiber optics ones) to solve this problem.

2) Discussion does not provide any data, comparing the characteristics of the sensor with the designs of other authors. In this context, even the statement about easy fabrication and good repeatability looks unfounded.

3) The y-axis in figure 2 is not labeled

I suggest to finalize the article so as not to lose such good results for publication.

english is fine

Reviewer 2 Report

Simultaneous detection of curvature and strain using a cascade interferometric structure is experimentally demonstrated by Shaocui Jiang et al. The authors fabricated an interferometric device based on the Mach-Zhender interferometer and an Anti-resonant cavity. The structure and the parameter detection are interesting; however, there are some major concerns related to terminology, sensitivity, and purpose of the designed structure that made it hard to recommend the acceptance of the manuscript in Photonics; the comments are below:

*The discussion about the principle of operation is suitable. However, the term “core offset” has a different meaning. The optical fiber sensors based on core-offset employ lateral misaligning splice, not in mismatch core splice. The authors should modify this part. 

*The sensitivities could be more impressive. The authors must include a comparative table regarding sensitivity, interferometric fiber structure, and publication year. 

*Independent strain and temperature sensitivity analysis of MZI and AR interferometric structure needs discussion and analysis. Then,  it is necessary to justify the cascade configuration using MZI and AR. If the sensitivities are higher than the cascade configuration, what is the reason for proposing this configuration?

Reviewer 3 Report

1.       Fig. 2 is the simulation results ? But the necessary parameters of simulation is not given.

2.       The fabrication process in Fig. 1 should be demonstrated in Section 3, not in Introduction.

3.       A four-mode fiber is chosen in this paper, but why only two modes are used to interfere ?

4.       The crosstalk should be discussed after Eq. (14).

5.       Why the length of FMF is 30 cm ? It is not a small value for a typical modal interferemeter.

6.       The references are not enough, especially for the strain, the following articles are advised to be referenced.

(1) The tapered fiber for axial strain sensing

(2) The micro-cavity structure for axial-strain sensing

No

Round 2

Reviewer 1 Report

The authors did a good job of finalizing the manuscript, even carried out the missing experiments

To be fair, [38] was the
first reference that came up just to illustrate the technique.
It is not the article where the technique was invented, so it is not necessary to cite this paper[38].

Up to me, english is acceptable

Reviewer 2 Report

It is a pleasant surprise the improvement of the manuscript; the authors addressed and clarified most of my points; In the present form, the manuscript is suitable for publication in Photonics. 

Reviewer 3 Report

This corrected manuscript is satisfied and can be accepted by this journal in the current form.